# Physiotherapy Rehabilitation in Subjects Diagnosed with Subacromial Impingement Syndrome Does Not Normalize Periscapular and Rotator Cuff Muscle Onset Time of Activation

**DOI:** 10.3390/ijerph18178952

**Published:** 2021-08-25

**Authors:** Silvia Ortega-Cebrián, Monserrat Girabent-Farrés, Rodney Whiteley, Caritat Bagur-Calafat

**Affiliations:** 1Physiotherapy Department, Universitat Internacional de Catalunya, 08195 Sant Cugat del Vallès, Spain; cbagur@uic.es; 2Rehabilitation Department, Aspetar Qatar Sports Medicine Hospital, Sport City Street, Doha 29222, Qatar; Rodney.Whiteley@aspetar.com; 3Department of Physiotherapy, School of Health Sciences, TecnoCampus-Pompeu Fabra University, 08302 Mataró, Spain; mgirabent@tecnocampus.cat

**Keywords:** onset time of activation, subacromial impingement, rehabilitation, electromyography

## Abstract

Clinicians suggest that rehabilitation of Subacromial Impingement Syndrome (SIS) should target improving movement patterns to ensure better clinical outcomes. Understanding changes in onset time of activation patterns and associated changes in clinical outcomes could improve our understanding of rehabilitation strategies. In this prospective longitudinal study, we examined neuromuscular firing patterns and clinical features before and after a standardized physiotherapy program in subjects diagnosed with SIS. Electromyography (EMG) recordings of eleven shoulder muscles were taken at the initial and discharge consultation in 34 male volunteers diagnosed with SIS. EMG recording was performed during flexion, scaption, and abduction at slow, medium, and fast speeds with a loaded (3 kg) and unloaded arm, as well as rotational motion, rotational strength, pain, and shoulder function. Completion of standardized shoulder physiotherapy program for SIS resulted in improvements in clinical outcomes. Resulted showed inconsistent differences of onset time of activation mainly in some of the periscapular muscles for all movements. No differences were seen on the EMG recordings for rotator cuff muscles. Differences in range of motion, strength and function were shown. Despite some changes in onset time of activation, this study was not able to demonstrate consistent changes of onset time of activation of the periscapular and rotator cuff muscles.

## 1. Introduction

The precise patho-anatomy of Subacromial Impingement Syndrome (SIS) is not fully understood. In 1867 the notion of mechanical irritation of the subacromial contents [including the subacromial bursa and the supraspinatus tendon] by the acromion was originally [1] and later popularized along with a proposed surgical management [2]. More recently our understanding has evolved to considering SIS not as a unique pathology but rather as multiple underlying pathologies [3,4,5]. SIS manifests as a variable presentation which can include a number of features including: pain; reductions of range of movement (ROM), strength, motor control, and function [6]. Shoulder function requires the interaction of coordinated movement [primarily glenohumeral and scapulothoracic joints], optimal muscle strength (axio-scapular, periscapular and rotator cuff) and an integrated sensorimotor system activity (proprioception and neuromuscular control) [7,8,9]. Accordingly, the presence of SIS could affect any of these systems [movement, muscles, sensorimotor], although it remains unclear whether they are a cause or the consequence of SIS. Rehabilitation of SIS is typically focused on decreasing pain, improving objective measures of ROM and strength, recovering altered movement and shoulder function [10]. Recovering altered movement seems more complex than the other clinical features, as motor control requires a sensorimotor systems that integrate peripheral sensation (afferent information through proprioceptive receptors), motor regulations (efferent response through neuromuscular control) and central pathways (processed at spinal level, brain stem, cerebral cortex and cerebellum) [11]. Clinically, motor control [afferent and efferent information] is indirectly assessed through the performance of functional tests, muscle strength tests, and/or muscle electromyography [12].

Altered movement patterns and poor functional performance are typically present in subjects diagnosed with SIS [13]. It is believed that pain and clinical features of SIS contribute to muscle inhibition and changes in neuromuscular patterns, decreasing the amount of muscle activation and delaying the onset of muscle activation [14,15,16]

Clinically, SIS rehabilitation programs integrate combinations of patient education, shoulder analytic and functional exercises, and passive care (e.g., manual therapy, modalities) to improve physical clinical outcomes and movement patterns [10,17,18]. It is suggested that resolution of altered movement is related to normalizing neuromuscular patterns - either muscle activation or onset time of activation. Electromyography (EMG) studies have demonstrated increased muscle activation and muscle strength after shoulder exercise programs in healthy subjects [10,19,20]. Conflicting results appear when onset time of activation is studied. In healthy participants, EMG studies in periscapular muscles have shown an initial activation of upper trapezius (UT), followed by serratus anterior (SA) and lower trapezius (LT) [18,21,22]. EMG shoulder muscle studies have showed greater variability in onset timing activation in subjects diagnosed with SIS [19,22]. Earlier activation of UT and delayed onset of SA and LT were reported [23,24,25], although the interplay between prime movers and rotator cuff when comparing SIS and healthy subjects have not shown differences in activation onset timing [26].

Despite the lack of normalized onset time of activation patterns in healthy and subjects diagnosed with SIS, rehabilitation programs are focused on restoring altered movement through different exercises [17]. Surprisingly, there are no data describing onset time of activation during the course of rehabilitation. Studying changes in onset time of activation patterns after rehabilitation could improve our understanding of the impact of SIS rehabilitation programs. Accordingly, this study aims to identify changes of onset time of activation of the prime movers and rotator cuff muscles of patients diagnosed with SIS after a standardized shoulder physiotherapy program.

## 2. Materials and Methods

In this prospective longitudinal study, onset time of activation was measured before and after a standardized physiotherapy rehabilitation program.

### 2.1. Settings and Participants

Forty-two male volunteers diagnosed with SIS were recruited and referred for the study by the sports medicine physicians at Aspetar Sports Medicine Hospital, Qatar. Participants met the criteria for inclusion if they were medically diagnosed with SIS and reported: a minimum of 3 months of mechanical shoulder pain; gradual onset of pain; pain aggravated during repetitive movements, or elevating the arm; reduced ROM by at least one third of the total range of motion (TROM); reduced rotatory strength; reduced shoulder function. On examination they displayed at least 3 of 5 positive impingement tests: Hawkins and Kennedy test, Neer test, empty can, painful arc, and Lift–off arm test [4,27,28,29]. Subjects requiring medical treatment for their shoulder in the previous year, or who had previous history of shoulder fractures, surgery, luxation, infections, labral injuries, shoulder instability, or apprehension to needles were excluded from the study (See Appendix B). Of the initial eligible 42 patients, 34 were included, volunteers received verbal and written information of the procedure, ethical principles followed the declaration of Helsinki, ethics approval was obtained from the local authority (Anti-Doping Lab Doha, Qatar, approval no: F2014000050) and participants provided signed informed consent prior to inclusion.

### 2.2. Variables and Measurements

The main variable of the study was onset time of activation (milliseconds, ms) of the periscapular muscles: upper trapezius (UT), lower trapezius (LT), anterior deltoid (AD), middle deltoid (MD), posterior deltoid (PD), pectoralis major (Pec) and serratus anterior (SA), and the rotator cuff muscles: infraspinatus (IF), supraspinatus (SS) and subscapularis (Subsc). These muscles were tested during flexion, scaption, and abduction at fast, mid, and slow speeds with both an unloaded and loaded arm at the initial and discharge physiotherapy consultations. The rehabilitation intervention comprised an individualized physiotherapy approach targeting the patient’s clinical deficits identified during examination. Each participant received treatment based on the individual’s standardized assessment which included pain, ROM, strength, and function as principal targets for improvement (See Appendix C). Further variables such as rotational ROM, rotational strength, pain and shoulder function were recorded as clinical outcome measures to describe the progress of rehabilitation.

After the first standardized shoulder physiotherapy assessment, all subjects underwent a tailored shoulder physiotherapy treatment three times a week for 12 weeks, based on individual daily assessment and treatment following a SIS physiotherapy treatment approach [23]. Pain reduction was addressed with isometric exercises and active gentle exercises, decreased ROM was addressed through accessory movements with manual therapy, manual stretching, and soft tissue techniques, decreased strength was treated with strength and resistance training, and decreased function was addressed using individual functional exercises, motor control education exercises, and conjunct training (10,23,29) (See Appendix C). Progression criteria was allowed when the patient reported pain at no more than 3/10 on the numeric rating scale (NRS). Weekly load progression was set at a 15% increase of exercise load performed the previous week [30].

### 2.3. Testing Procedure and Instrumentation

Electromyography (Delsys Inc., Boston, MA, USA) recordings were taken in the injured arm, using superficial EMG (sEMG) and indwelling EMG (iEMG) of the periscapular (UT, AD, MD, PD, LT Pec, and SA) and rotator cuff (IF, SS, Subsc) muscles during flexion, scaption, and abduction; loaded and unloaded; at slow, medium, and fast speeds (18 conditions). Superficial EMG recording were taken from UT, LT, AD, MD, PD, SA, Pec, while iEMG recorded IF, SS and Subsc. Superficial electrodes were applied following the SENIAM guidelines (http://www.seniam.org/ (accessed on 24 June 2016)) for skin preparation and localization. The introduction of indwelling EMG followed procedures previously documented for rotator cuff muscles [31,32] using a sterile technique with 30 mm hypodermic needles for SS and IF muscles and 50 mm for Subsc.

The researcher taught the specific movements to each participant before collecting the data. Movements were performed when standing with the elbow extended and the forearm in neutral. Each movement was performed three times with 5 seconds’ rest between repetitions. During the “slow” movement full elevation was performed in 5 s, “medium” in 3 s, and “fast” in 1 s. Movements were supervised and guided by the researcher using a stopwatch. When administering the “Load” condition (L), the researcher used a 3 kg dumbbell, and when administering the “unloaded” (UL) condition no additional weight was used.

Clinical outcome measures were obtained during a physiotherapy examination and consisted of recording levels of pain (NRS), and the levels of function using the patient specific functional scale questionnaire (PSFS). Physiological measures were also taken, consisting of rotational ROM performed in supine position at 90°, shoulder abduction at 90°, and elbow flexion with an inclinometer (Empire Magnetic Inclinometer magnetic, Empire Level Manufacturing Co, Milwaukee, WI, USA). Isometric internal and external rotation strength were recorded in standing position with the arm by the side and 90° of elbow flexion with a handheld dynamometer (HHD) (Ergofet500, Hoggan Scientific LLC, Salt Lake City, UT, USA).

### 2.4. Sample Size

As there are no comparable previous research findings available to allow a priori calculation of appropriate sample size estimations, we note that research documenting sEMG and iEMG in populations diagnosed with SIS have had varying sample sizes, typically approximately 30 participants [33]. Accordingly, the research was terminated after 34 were included. A post-hoc power analysis was conducted to establish the veracity of this approach.

### 2.5. Signal Processing and Data Analysis

Prior to data acquisition, an EMG signal test and voluntary contractions were performed to confirm correct electrode placement. All EMG data were collected with a 16-bit digital system, with a gain of 300. The signal was band pass filtered at 5 Hz-200 Hz and converted from analogue to digital at 2000 Hz. The band pass filter was tested by the investigators to ensure minimal noise for iEMG. Data analyses were performed using custom methods written in MATLAB R2009a (MATLAB & Simulink, MathWorks, Inc., Reykjavik, Iceland). The EMG signal was normalized relative to the resting baseline, which was the EMG recording while the subject was instructed to relax with their arm by the side [32]. Timing of activation of the muscles was normalized to onset of arm movement, which was detected by an accelerometer (accelerometer sample rate: 250 Hz, Delsys, Inc., Boston, MA, USA) attached to the subject’s wrist.

In order to determine successful treatment at the end of 12 weeks treatment, participants had to demonstrate increased rotational ROM and rotational strength, reduced pain levels and recovery of complete shoulder function.

Data were initially examined for normality, and descriptive statistics calculated. To identify differences of onset, time of activation of each muscle for each movement (flexion, scaption, abduction), at each speed (slow, medium, fast) and load (loaded and unloaded) before and after the standardized shoulder physiotherapy program for SIS Wilcoxon signed-rank test was calculated. To determine successful treatment for the study, statistically significant differences before and after treatment for pain, function (PSFS), rotational ROM, and strength were calculated with a dependent paired *t*-test. Hedges’ g was used to calculate the effect size, as the prior sample data differs from the after sample data. Statistical significance of an alpha level of 0.05 was applied. All the statistical calculations were performed with SPSS 21.0 (IBM, Armonk, NY, USA) software.

## 3. Results

Thirty-four patients were examined prior to commencing rehabilitation. Eight volunteers were excluded from the study because rehabilitation attendance was less than 50%. Twenty six of the 34 initial volunteers were analyzed after rehabilitation (Table 1). Descriptive statistics for onset time of activation (ms) median and interquartile range (IQR) are summarized for each muscle, movement, speed and load in Appendix A.

### 3.1. Flexion Movement

Onset time of activation during flexion showed statistically significant differences in loaded slow speed for UT (*p* = 0.01), AD (*p* = 0.00) and SA (*p* = 0.05) (Table 2). During flexion loaded at medium speed MD (*p* = 0.00), PD (*p* = 0.02), LT (*p* = 0.02) and Pec (*p* = 0.05) were different after treatment, while for loaded and fast speed only Pec showed differences after treatment. Fewer differences were seen in the unloaded condition, whereas in slow speed only AD (*p* = 0.00) showed a difference. In unloaded medium speed, UT (*p* = 0.00) and AD (*p* = 0.00) were statistically significant different after treatment, as well as in unloaded fast speed UT (*p* = 0.00), and PD (*p* = 0.05) also showed a difference.

### 3.2. Scaption Movement

Results of scaption showed statistically significant differences in loaded slow speed for UT (*p* = 0.03), AD (*p* = 0.00), MD (*p* = 0.00) and PD (*p* = 0.05). Loaded medium speed MD (*p* = 0.00), and PD (*p* = 0.00) showed a difference after treatment, but no differences were seen in loaded fast speed. Regarding the unloaded conditions, differences occurred in all three speeds, but only MD (*p* = 0.01) showed difference in slow speed. In unloaded medium speed UT (*p* = 0.00) and AD (*p* = 0.00) were different after treatment, while in unloaded fast speed UT (*p* = 0.00) and MD (*p* = 0.05) showed statistically significant difference after treatment (Table 3).

### 3.3. Abduction Movement

Onset time of activation during abduction showed statistically significant differences in all conditions of load and speed. During loaded abduction in slow speed, AD (*p* = 0.00) and MD (*p* = 0.01) were different after treatment. AD (*p* = 0.00), PD (*p* = 0.05) and LT (*p* = 0.01) showed differences in medium speed, whereas in fast speed AD (*p* = 0.00) and MD (*p* = 0.04) demonstrated significant difference after treatment. In unloaded condition, statistically significant differences were seen in UT (*p* = 0.02) and MD (*p* = 0.02) in slow speed, UT (*p* = 0.04), AD (*p* = 0.01), MD (*p* = 0.01) and PD (*p* = 0.02) in medium speed and AD (*p* = 0.04), and SA (*p* = 0.03) in fast speed (Table 4).

While onset time of activation during flexion, scaption and abduction showed statistically significance mainly in periscapular muscles, RC muscles did not show difference in onset time of activation after standardized physiotherapy treatment.

The study has not been able to demonstrate a tendency towards faster or slower onset time of activation after a standardized shoulder physiotherapy program for SIS; positive values of effect sizes are associated with faster onset time of activation, while negative values imply slower onset times of activations. In Figure 1, Figure 2 and Figure 3, differences of onset time of activation are shown.

Significant improvements in measurements of rotational ROM, strength, pain, and function were seen after standardized physiotherapy treatment. Specifically, improvements in rotational movement (IR 19.07°, *p* = 0.000; ER 8.18°, *p* = 0.000; IR at 90° of Flex 15.3°, *p* = 0.000), strength levels (IR 5.2 Kg, *p* = 0.000; ER 6.6 Kg, *p* = 0.000) and function (23.07 PSFS) were demonstrated. Pain was not completely resolved (1.4/10) after treatment (see Table 5).

### 3.4. Post-Hoc Sample Size Calculation

A post-hoc power-analysis estimated that, using a power of 70% and α = 0.05, 19 patients would have been needed to show a significant change of onset time of activation for periscapular muscles (Hedges’ g = 0.65), while 42 participants would be adequate in the case of rotator cuff muscles (Hedges’ g = 0.37).

## 4. Discussion

Despite significant clinical improvements across the duration of rehabilitation, no changes in the onset timing for the rotator cuff muscles were observed during this study. These findings suggest that timing of activation of the periscapular and rotator cuff muscles is not well related to changes in clinical features, and therefore may not be an optimal rehabilitation target. In addition, we were unable to describe any pattern of delayed or anticipated onset time of activation after a rehabilitation program.

Inconsistent changes were documented for the periscapular muscles depending on the condition examined. Altered onset time or level of activation is related to abnormal shoulder neuromuscular patterns, (e.g., scapular dyskinesia) which is associated with shoulder dysfunction in the presence of SIS [35]. In SIS, rehabilitation is focused on normalizing neuromuscular patterns and improving clinical features [25,26,36]. Clinically, it is believed that neuromuscular patterns are normalized once pain, strength, function, and aberrant shoulder movement patterns are resolved, despite the lack of specific information regarding onset time of activation after physiotherapy programs [18,25,37]. This study is the one of the few investigation describing onset time of activation of periscapular and rotator cuff muscles before and after completing a rehabilitation program. Evidence exists of earlier activation of periscapular muscles compared to rotator cuff muscles in healthy subjects in single EMG recording studies, [25,26,38,39], but of no differences when studying subjects diagnosed with SIS [26]. Although previous research suggested that onset time of activation patterns during physiological movements are not able to discriminate pathology [26], delayed onset time of activation in shoulder muscles has been related to altered movement patterns and dysfunction [15,40] despite the difficulty in describing an onset time of activation pattern in patients diagnosed with SIS and healthy subjects [9,41]. Evidence for efficacy of the concept of normalizing onset activation patterns after shoulder physiotherapy programs seems sparse and the current data questions the appropriateness of directing attention to shoulder muscle firing patterns during rehabilitation.

This study demonstrates changes in onset activation patterns mainly in periscapular muscles during physiological movements. In flexion, differences were seen at loaded slow (UT, AD and SA) and medium speeds (MD, PD and LT), while in the unloaded condition differences were seen at slow (AD and Pec), medium (UT, AD, Pec) and fast speeds (UT, AD and PD). Earlier activation of AD and SA have been reported in healthy subjects during flexion [22], as well as SS and IF in loaded and unloaded conditions [22]. Worsley reported improvements in delayed activation of SA and UT after 10 weeks’ scapula muscle training, although there remained a delay compared to healthy comparison group [18].

During scaption, changes in onset time of activation were seen at the loaded slow (UT, AD, MD and PD) and medium speeds (MD and PD), while at the unloaded slow condition (MD), medium speed (UT and AD) and fast speed (UT, MD and SA) saw changes. Previous research has not identified changes in time of activation during elevation in the scapular plane after rehabilitation [19]. In contrast, experimental studies have demonstrated changes in onset time of activation in UT, SA, and LT after performing dry needling, although the study did not describe the direction of these changes [41].

Abduction demonstrated differences in onset time of activation at loaded slow (AD and MD), medium (AD, PD and LT), and fast speed (AD, and MD), whereas at the unloaded condition slow (UT and MD), medium (UT, AD, MD and PD) and fast speed (AD and SA) showed differences. Again, the current research conflicts with assertions that that onset time of activation can be delayed after motor control training [18].

Previous studies have suggested a delayed onset of the RC muscles as the reason for pain and dysfunction [17]. The present study does not support this theory as there were no differences observed in onset time of activation of the RC muscle despite changes in pain and function after rehabilitation of SIS [10,20]. Clinically, the relevance of the findings of this study suggest that rehabilitation programs should target restoration of ROM, strength and function rather than changes in onset time of activation. Furthermore, the notion of the “stabilizing” function of RC muscles during gross movements is also questioned, as onset time of activation of the RC muscles seems to occur before movement at slow and medium (but not fast) speeds, and unloaded but not loaded conditions. These results would suggest that, clinically, a “stabilizing” function of the rotator cuff muscles does not occur universally as we had suspected. In addition, EMG of the RC muscles showed essentially unpredictable onset time of activation despite the direction of rotation not showing a “direction specific” pattern [22,42], and these findings suggests that clinicians could prescribe any rotational direction movements to activate RC. Again, the more striking changes in ROM, strength, and function suggest that those are the clinical features to target in rehabilitation programs for SIS rather than normalizing onset time of activation.

The results of this study support previous research where clinical features of subjects diagnosed with SIS demonstrated moderate to large increases in ROM of IR, ER, and in IR at 90°of Flexion (Hedge’s g: −0.61, −0.33, −0.50, respectively) after successful treatment [35]. A larger effect was seen for IR (g: −1.06) and ER (g: −0.51) strength, function (g: −1.84) and pain (g: 2.23) in the current study.

Limitations of the current study include lost data, mainly of the RC muscles due to technical issues, the combination of sEMG and iEMG [25], and the large variability of EMG data. Larger sample sizes would counter these difficulties, and the present data allow for a priori power analysis should future research attempt to re-address these questions [43]. Further shoulder EMG studies would be needed to describe the direction of changes associated with rehabilitation, and delayed onset of activation is associated with symptoms. It is noted that the differences in activation for the different muscle groups occur in the order of approximately 0.01 to 0.2 seconds. It remains to be seen if these changes are able to be measured in a clinical setting (without EMG) in the context of rehabilitation programs. Before undertaking further research in this area, it is suggested that documenting if it is even possible for patients to be capable of reliably performing such fine motor activation changes would be a sensible first step.

## 5. Conclusions

Changes of onset time of activation occur mainly in periscapular muscles after completing a standardized shoulder physiotherapy program for SIS. Upper trapezius, anterior deltoid and mid deltoid presented most of the changes in onset time of activation during all movement (flexion, scaption and abduction), speeds (fast, medium and slow), and loads (loaded and unloaded) condition. The unloaded condition demonstrated greater changes than the loaded condition in all movements and rotator cuff muscles did not show changes in onset time of activation. No systematic changes in activation timing (earlier or later) were discernable after successfully completing a standardized shoulder physiotherapy program for SIS.

Although completion of standardized shoulder physiotherapy program for SIS resulted in improvements in rotational ROM, strength, function, and pain, these improvements were not meaningfully related to changes in onset time of activation patterns of the 11 muscles examined during the different movements at different speeds, in loaded and unloaded conditions. The authors suggest that changes in onset time of activation appear to be independent of clinical improvements. Thus, onset time of activation does not represent a sensible clinical target for rehabilitation, suggesting the targeting of clinical outcomes in decreasing pain and increasing ROM, strength and function.

## Figures and Tables

**Figure 1 ijerph-18-08952-f001:**
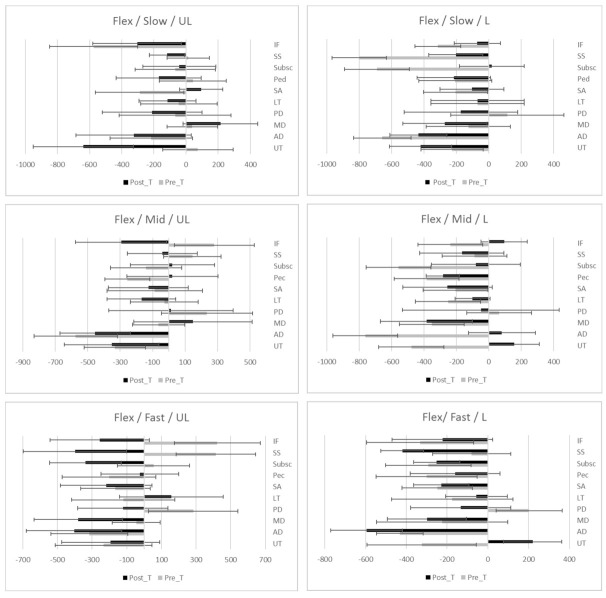
EMG onset time (ms) of activation (median ± IQR) during Flexion prior and after standardized physiotherapy treatment for SIS.Descriptive data: Median, IQR (Inter Quartile Range) of onset time of activation during flexion movement at different load and speed. UL (Unloaded), L (Loaded), Slow (slow speed), Mid (medium speed) Fast (fast speed). Muscles; UT (Upper Trapezius), AD (Anterior Deltoid), MD (Mid Deltoid), PD (Posterior Deltoid), LT (Lower Trapezius), SA (Serratus Anterior), Pec (Pectoralis Major), Susbc (Subscapularis), SS (Supraspinatus), IF (infraspinatus).

**Figure 2 ijerph-18-08952-f002:**
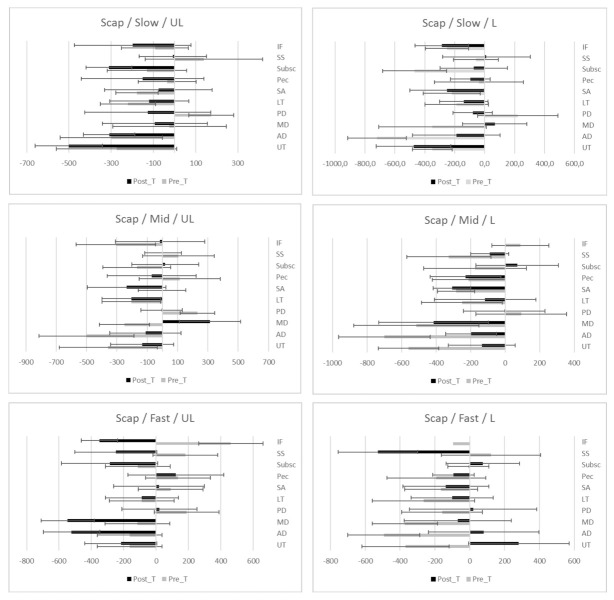
EMG onset time (ms) of activation (median ± IQR) during Scaption prior and after standardized physiotherapy treatment for SIS. Descriptive data: Median, IQR (Inter Quartile Range) of onset time of activation during flexion movement at different load and speed. UL (Unloaded), L (Loaded), Slow (slow speed), Mid (medium speed) Fast (fast speed). Muscles; UT (Upper Trapezius), AD (Anterior Deltoid), MD (Mid Deltoid), PD (Posterior Deltoid), LT (Lower Trapezius), SA (Serratus Anterior), Pec (Pectoralis Major), Susbc (Subscapularis), SS (Supraspinatus), IF (infraspinatus).

**Figure 3 ijerph-18-08952-f003:**
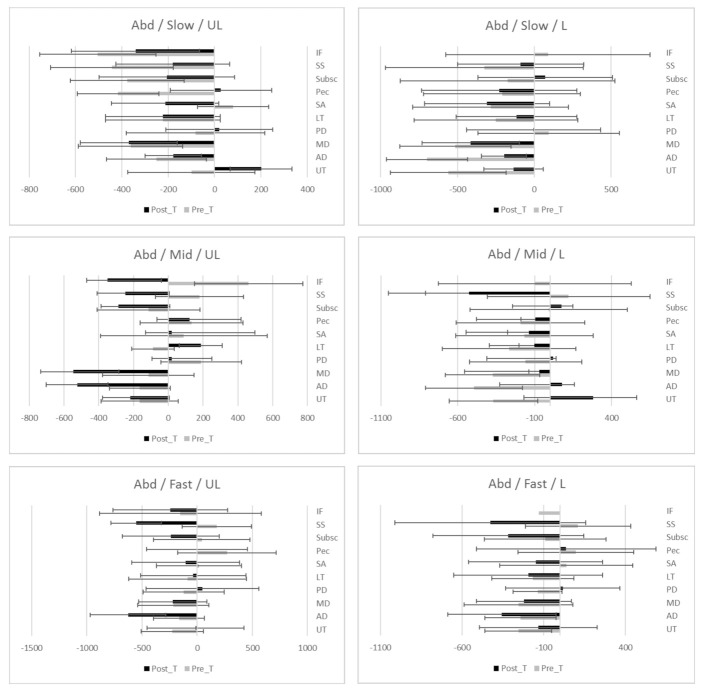
EMG onset time (ms) of activation (median ± IQR) during Abduction before and after standardized physiotherapy treatment for SIS. Descriptive data: Median, IQR (Inter Quartile Range) of onset time of activation during flexion movement at different load and speed. UL (Unloaded), L (Loaded), Slow (slow speed), Mid (medium speed) Fast (fast speed). Muscles; UT (Upper Trapezius), AD (Anterior Deltoid), MD (Mid Deltoid), PD (Posterior Deltoid), LT (Lower Trapezius), SA (Serratus Anterior), Pec (Pectoralis Major), Susbc (Subscapularis), SS (Supraspinatus), IF (infraspinatus).

**Table 1 ijerph-18-08952-t001:** Demographic data of study participants.

Demographics	Pre_Treatment	Post_ Treatment
	Mean [SD]	
**Age [years]**	29.1 [5.5]	28.4 [2.1]
**Height [cm]**	178 [9.5]	177 [3]
**Weight [Kg]**	78.2 [8.7]	79.1 [7.6]
**Affected Side**	%	
R	83%	87%
L	17%	13%
**Physical Activity ***	[n]	
0–3	0	0
4–7	3	2
8–10	31	24

Demographic data; cm (centimeters), Kg (Kilograms), SD (Standard Deviation), R (Right side), L (Left side) * Tegner Scale for physical activity [34].

**Table 2 ijerph-18-08952-t002:** Comparative statistics of EMG onset time of activation (ms) during flexion before and after standardized physiotherapy treatment for SIS movement at slow, medium and fast speeds; loaded and unloaded conditions.

Speed	Muscle	Flexion LOADED				Flexion UNLOADED			
		Hedges’s g Effect Size	95% CI Sup Lim	95% CI Inf Lim	*p*-Value	Hedges’s g Effect Size	95% CI Sup Lim	95% CI Inf Lim	*p*-Value
**Slow**	**UT**	1.97	243.58	−96.73	0.01	3.08	−30.74	−424.60	0.37
	**AD**	0.33	−51.00	−381.30	0.00	3.11	−506.23	−807.97	0.00
	**MD**	−0.35	224.31	−146.96	0.07	0.35	41.59	−294.28	0.53
	**PD**	0.44	372.24	−507.82	0.68	0.62	399.07	−172.41	0.33
	**LT**	0.20	293.93	−384.34	0.72	0.92	164.51	−304.01	0.58
	**SA**	−0.73	236.14	−812.27	0.05	0.16	44.61	−455.55	0.43
	**Pec**	−0.05	243.94	−375.96	0.11	5.75	−485.13	−896.82	0.07
	**Subsc**	0.51	306.96	−222.74	0.79	1.08	9.89	−423.90	0.45
	**SS**	0.38	500.48	−471.55	0.18	0.72	−638.30	−960.74	0.18
	**IF**	−0.69	−117.60	0.38	0.34	2.85	125.11	−755.86	0.07
**Medium**	**UT**	0.06	−191.31	−475.17	0.26	3.38	−330.17	−626.43	0.00
	**AD**	−0.39	−457.93	−691.93	0.21	5.94	−657.01	−865.40	0.00
	**MD**	−0.53	69.68	−195.54	0.00	0.40	−250.48	−450.86	0.44
	**PD**	0.62	407.07	64.14	0.02	2.73	267.85	−141.67	0.44
	**LT**	0.36	164.40	−220.02	0.02	0.16	−35.26	−468.21	0.65
	**SA**	0.12	135.42	−308.99	0.65	0.47	3.59	−415.05	0.88
	**Pec**	−0.60	−74.37	−435.43	0.05	1.43	−176.90	−590.82	0.06
	**Subsc**	−0.37	204.64	−485.64	0.11	1.23	−295.80	−815.91	0.40
	**SS**	0.39	415.42	−124.71	0.22	3.59	262.48	−440.58	0.59
	**IF**	1.93	634.81	−73.13	0.78	0.07	113.23	−585.61	0.91
**Fast**	**UT**	−0.11	−133.38	−333.97	0.59	4.62	−191.86	−457.84	0.00
	**AD**	0.20	−217.80	−413.58	0.59	0.68	−319.26	−543.20	0.09
	**MD**	0.92	45.96	−136.59	0.20	0.14	−109.47	−338.57	0.39
	**PD**	0.85	421.60	145.51	0.53	2.00	346.78	58.02	0.05
	**LT**	−0.55	21.82	−261.87	0.69	0.48	−30.66	−318.45	0.72
	**SA**	0.12	−64.88	−267.09	0.86	0.06	−110.84	−385.73	0.90
	**Pec**	−0.38	−45.61	−358.90	0.05	1.15	−135.25	−465.33	0.79
	**Subsc**	0.77	329.02	−219.99	0.59	1.15	24.74	−610.74	0.18
	**SS**	2.03	614.74	213.51	0.14	2.60	240.71	−399.26	0.65
	**IF**	1.51	634.52	210.54	0.76	0.18	−28.50	−636.61	0.56

*p* > 0.05, Hedges‘ Effect Size, 95% Confidence Interval (CI) Superior Limit (Sup Lim) and Inferior Limit (Inf Lim), and p-value of comparative EMG onset time of activation of flexion movement at different speeds (slow, medium, fast) and load (loaded and unloaded). Non-parametric data. UT (Upper Trapezius), AD (Anterior Deltoid), MD (Mid Deltoid), PD (Posterior Deltoid), LT (Lower Trapezius), SA (Serratus Anterior), Pec (Pectoralis Major), Susbc (Subscapularis), SS (Supraspinatus), IF (infraspinatus).

**Table 3 ijerph-18-08952-t003:** Comparative statistics of EMG onset time of activation (ms) during scaption movement prior to and after standardized physiotherapy treatment for SIS movement at slow, medium, and fast speeds; loaded and unloaded conditions.

Speed	Muscle	Scaption LOADED				Scaption UNLOADED			
		Hedges’s g Effect Size	95% CI Sup Lim	95% CI Inf Lim	*p*-Value	Hedges’s g Effect Size	95% CI Sup Lim	95% CI Inf Lim	*p*-Value
**Slow**	**UT**	0.60	−81.23	−468.09	0.03	0.31	−186.93	−513.17	0.74
	**AD**	0.04	−133.55	−465.09	0.00	5.40	−615.21	−823.03	0.77
	**MD**	0.22	155.91	−203.27	0.00	5.85	−198.25	−499.14	0.01
	**PD**	0.67	412.94	−65.66	0.05	0.48	443.25	1.34	0.11
	**LT**	−0.23	0.77	−440.89	0.66	0.39	36.29	−410.56	0.88
	**SA**	−0.19	−212.7	531.7	0.75	0.05	−10.31	−429.51	0.46
	**Pec**	0.28	226.47	−297.15	0.12	0.43	174.11	−250.25	0.78
	**Subsc**	0.37	251.52	−512.41	0.35	0.46	−219.16	−719.62	0.89
	**SS**	0.34	450.91	−171.47	0.65	7.59	520.09	−636.19	0.89
	**IF**	0.26	550.06	−735.56	0.56	1.27	285.45	−788.83	0.09
**Medium**	**UT**	−0.82	−238.30	−476.33	0.10	1.41	−418.91	−700.89	0.00
	**AD**	−1.09	−387.98	−610.87	0.99	2.24	−604.29	−793.14	0.00
	**MD**	−1.30	−117.75	−382.60	0.00	0.64	−382.35	−645.01	0.19
	**PD**	0.58	399.19	62.49	0.00	0.85	277.01	−90.69	0.72
	**LT**	0.00	−3.80	−405.18	0.72	0.50	−45.79	−453.84	0.33
	**SA**	0.57	−65.3	−404.6	0.30	0.47	−97.70	−471.59	0.16
	**Pec**	0.43	314.00	−84.76	0.23	0.15	−19.23	−402.52	0.37
	**Subsc**	−0.35	236.92	−574.35	0.72	0.23	203.98	−550.98	0.87
	**SS**	−0.89	149.30	−762.87	−0.55				
	**IF**	0.24	437.49	−224.71	0.18	0.09	71.81	−723.41	0.65
**Fast**	**UT**	0.18	−33.49	−295.51	0.64	5.22	−263.87	−472.63	0.00
	**AD**	1.09	−28.20	−299.16	0.89	8.06	−379.26	−608.43	0.52
	**MD**	1.10	25.62	−254.84	0.91	4.94	−262.05	−484.09	0.05
	**PD**	0.32	338.62	39.16	0.83	1.66	−21.38	−295.84	0.66
	**LT**	0.00	77.69	−254.02	1.00	1.35	−105.10	−425.37	0.07
	**SA**	0.14	119.7	−331.1	1.00	0.42	0.37	−329.44	0.02
	**Pec**	0.03	277.43	−7.34	0.35	1.07	−34.64	−348.51	0.21
	**Subsc**	0.35	211.24	−438.56	0.65	0.16	220.14	−236.64	0.69
	**SS**	0.79	375.25	−15.59	0.08	0.01	387.78	−144.20	0.24
	**IF**	1.91	681.00	242.95	−0.61				

*p* > 0.05, Hedge‘s Effect Size, 95% Confidence Interval (CI) Superior Limit (Sup Lim) and Inferior Limit (Inf Lim), and *p*-value of comparative EMG onset time of activation of scaption movement at different speeds (slow, medium, fast) and load (loaded and unloaded). Non-parametric data. UT (Upper Trapezius), AD (Anterior Deltoid), MD (Mid Deltoid), PD (Posterior Deltoid), LT (Lower Trapezius), SA (Serratus Anterior), Pec (Pectoralis Major), Susbc (Subscapularis), SS (Supraspinatus), IF (infraspinatus).

**Table 4 ijerph-18-08952-t004:** Comparative statistics of EMG onset time of activation (ms) during abduction before and after standardized physiotherapy treatment for SIS movement at slow, medium and fast speeds; loaded and unloaded conditions.

Speed	Muscle	Abduction LOADED				Abduction UNLOADED			
		Hedges’s g Effect Size	95% CI Sup Lim	95% CI Inf Lim	*p*-Value	Hedges’s g Effect Size	95% CI Sup Lim	95% CI Inf Lim	*p*-Value
**Slow**	**UT**	−0.74	81.61	−281.80	0.18	0.38	−166.55	−562.20	0.02
	**AD**	−0.34	−460.94	−725.36	0.00	−0.04	−94.91	−406.38	0.37
	**MD**	0.35	−196.33	−528.36	0.01	0.87	−427.25	−703.21	0.02
	**PD**	−0.24	155.77	−320.73	0.98	2.33	−60.46	−440.32	0.69
	**LT**	0.00	93.61	−538.11	0.92	1.56	247.04	−228.33	1.00
	**SA**	0.60	−36.8	477.7	0.22	0.75	−55.81	−458.80	0.39
	**Pec**	−0.92	−107.60	−726.33	0.13	−0.19	345.71	−270.93	0.21
	**Subsc**	−0.34	−0.05	−752.03	0.11	1.08	300.60	−260.25	0.60
	**SS**	−0.61	−52.37	−833.87	0.11	8.02	112.64	−671.91	0.65
	**IF**					3.05	247.2	−261.9	0.09
**Medium**	**UT**	0.46	−126.72	−403.28	0.44	0.01	−397.59	−662.30	0.04
	**AD**	−0.86	−517.35	−754.55	0.00	1.37	−240.20	−489.40	0.01
	**MD**	−1.18	−308.06	−557.18	0.09	−0.19	−497.22	−744.51	0.01
	**PD**	−0.67	−60.95	−391.66	0.05	0.63	−118.99	−448.76	0.02
	**LT**	0.00	158.87	−324.17	0.01	0.06	243.00	−159.20	0.36
	**SA**	0.27	80.4	554.1	0.42	0.08	−35.62	−429.65	0.30
	**Pec**	0.39	315.75	−110.53	0.06	−0.35	328.09	−236.08	0.08
	**Subsc**	−0.12	195.30	−497.48	1.00	0.06	296.63	−394.43	0.59
	**SS**	0.76	297.10	−404.20	0.14	4.11	297.10	−404.20	0.67
	**IF**								
**Fast**	**UT**	−0.61	−124.09	−332.30	0.28	1.62	−124.69	−385.77	0.43
	**AD**	−0.70	−119.55	−364.55	0.00	−0.74	−79.90	−251.20	0.04
	**MD**	0.01	−99.16	−333.90	0.04	1.48	−148.94	−363.17	0.69
	**PD**	−0.31	15.38	−261.90	0.93	0.21	−3.95	−271.96	0.57
	**LT**	0.00	116.06	−290.59	0.17	1.14	216.20	−0.04	0.92
	**SA**	0.22	15.4	−261.9	0.88	0.81	195.40	−118.82	0.03
	**Pec**	0.55	444.55	95.92	0.50	0.45	315.09	−123.54	0.47
	**Subsc**	0.74	298.86	−215.43	0.59	0.03	158.55	−342.04	0.72
	**SS**	3.04	340.88	13.58	0.12	1.19	414.72	−194.27	0.89
	**IF**					0.16	277.57	−582.88	0.35

*p* > 0.05, Hedges’ Effect Size, 95% Confidence Interval (CI) Superior Limit (Sup Lim) and Inferior Limit (Inf Lim), and *p*-value of comparative EMG onset time of activation of abduction movement at different speeds (slow, medium, fast) and load (loaded and unloaded). Non-parametric data. UT (Upper Trapezius), AD (Anterior Deltoid), MD (Mid Deltoid), PD (Posterior Deltoid), LT (Lower Trapezius), SA (Serratus Anterior), Pec (Pectoralis Major), Susbc (Subscapularis), SS (Supraspinatus), IF (infraspinatus).

**Table 5 ijerph-18-08952-t005:** Descriptive statistics of clinical features of ROM, strength, pain and function.

Descriptive Data Physical Outcome Measures									
Variable	Condition	n	Mean	SD	Variance	95% IC Sup Lim	95% IC Inf Inf	*p*-Value	Hedges’s g Effect Size
**ROM IR**	**Pre_T**	34	33.03	13.03	26.1	39.3	169.8	0.00	−0.61
	**Post_T**	26	52.1	15.2	44.6	57.9	230.2		
**ROM ER**	**Pre_T**	34	96.3	1.31	88.8	99.2	169.4	0.00	−0.33
	**Post_T**	26	104.5	7.7	101.2	108.1	58.9		
**ROM IR 90°Flex**	**Pre_T**	34	23.5	9.2	18.7	28.1	84.9	0.00	−0.50
	**Post_T**	26	38.5	17.9	30.4	46.6	320.1		
**S Kg ER**	**Pre_T**	34	10.6	3.0	9.7	12.5	9.0	0.00	−0.51
	**Post_T**	26	17.2	2.4	16.1	18.2	5.7		
**S Kg IR**	**Pre_T**	34	18.4	4.8	17.2	21.4	22.7	0.00	−1.06
	**Post_T**	26	23.6	4.2	21.8	25.6	17.9		
**PSFS**	**Pre_T**	34	21	5.9	18.3	23.5	35.9	0.00	−1.84
	**Post_T**	26	44.1	5.0	41.7	46.1	25.3		
**NRS**	**Pre_T**	34	6.7	1	6.3	7.3	1.2	0.00	2.23
	**Post_T**	26	1.4	1.2	0.9	2.0	1.5		

Descriptive statistics of key clinical features, parametric data. ROM (degrees), Strength (Kg), NRS (Numeric Rate Score (pain) and PSPF (Patient Specific Functional Scale). ROM IR (Range of Motion Internal Rotation), ROM ER (Range of Motion External Rotation), ROM IR 90 Flex (Range of Motion at 90 Degrees of Internal Rotation), S Kg ER (strength Kilogram for External Rotation), S Kg IR (Strength Kilogram for Internal Rotation), Pre T (Previous to treatment) and Post T (Posterior to Treatment).

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
