# Peer review of "Physiotherapy Rehabilitation in Subjects Diagnosed with Subacromial Impingement Syndrome Does Not Normalize Periscapular and Rotator Cuff Muscle Onset Time of Activation"

_ijerph, 2021, doi:10.3390/ijerph18178952_

Round 1

Reviewer 1 Report

Surnames should not appear in the introduction. These parts of the introduction should be included in the discussion.

In the introduction, a sentence should be added that the shoulder is an anatomical area and subacromial space is a part of it. And what is the most important cause of SIS from a biomechanical point of view.

Materials and methods.

The inclusion criteria should be clearly stated, preferably in the form of a table.

Flow chart should be added.

The conclusions should be more concise and strictly correspond to the research hypothesis.

Author Response

Thank you for your comments,

  • Point 1: Surnames should not appear in the introduction. These parts of the introduction should be included in the discussion.

The introduction has been revised as suggested

  • Point 2: In the introduction, a sentence should be added that the shoulder is an anatomical area and subacromial space is a part of it. And what is the most important cause of SIS from a biomechanical point of view.

We thank the reviewer for the comment and the opportunity to expand and clarify the introduction as suggested.

  • Point 3: Materials and methods.
  • The inclusion criteria should be clearly stated, preferably in the form of a table.
  • Flow chart should be added.

We have added an inclusion criteria flow chart in the appendix (Appendix A in supplementary data).

  • Point 4: The conclusions should be more concise and strictly correspond to the research hypothesis.

We have rewritten the conclusion following your comments

Reviewer 2 Report

Hello!

This is a good study and the paper should be published but there are many grammatical and spelling errors. I’ve attempted to fix most of them here. Also your supplemental tables use commas instead of periods. p=.05, not p=0,05.  This must be fixed. See the attached file for grammatical assistance. Best wishes to you and your team.

Author Response

Thank you for the kind comments.

Point 1: This is a good study and the paper should be published but there are many grammatical and spelling errors. I’ve attempted to fix most of them here. Also your supplemental tables use commas instead of periods. p=.05, not p=0,05.  This must be fixed. See the attached file for grammatical assistance. Best wishes to you and your team.

All tables of the manuscript and supplementary data have been reviewed; commas have been changed for periods as requested, and a final grammatical review conducted.

Reviewer 3 Report

Thank you for giving me the possibility to review the manuscript "Physiotherapy rehabilitation in subjects diagnosed with Sub-acromial Impingement Syndrome does not normalize shoulder muscle onset time of activation. The manuscript aims to identify changes in the onset time of activation: prime movers and rotator cuff muscles of patients diagnosed with SIS after a standardized shoulder physiotherapy program during physiological movements.

However, before considering it for publication on IJERPH, the following issues should be addressed:

  • English grammar/spelling revision is mandatory;
  • please add a table summarizing the main features of the recruited volunteers (mean age; gender; affected side; medical comorbidities; job; physical activity level (according to Tegner scale)...
  • please conduct a gender-specific sub-analysis to assess if there is a between gender variability in the findings of the present study;
  • please discuss further the clinical relevance of the present study's findings and its potential future developments.

Author Response

Thank you for your comments,

Point 1: English grammar/spelling revision is mandatory;

Grammatical and spelling corrections have been done as requested by a native English speaker.

Point 2: please add a table summarizing the main features of the recruited volunteers (mean age; gender; affected side; medical comorbidities; job; physical activity level (according to Tegner scale)..

Table has been added in the paper (See table 1)

Point 3: please conduct a gender-specific sub-analysis to assess if there is a between gender variability in the findings of the present study;

All participants were male, therefore gender sub analysis cannot be performed. This information has now been clarified in the text : “Forty-two male volunteers diagnosed with…” in the 2.1. Settings and participants section”

Point 4: please discuss further the clinical relevance of the present study's findings and its potential future developments

We have added further clinical implications if the study’s findings.

Round 2

Reviewer 1 Report

As it stands, the article is much better and ready to print.
Best Regards